# INFERRING DYNAMICAL SYSTEMS WITH LONG-RANGE DEPENDENCIES THROUGH LINE ATTRACTOR REGULARIZATION

## ABSTRACT

Vanilla RNN with ReLU activation have a simple structure that is amenable to systematic dynamical systems analysis and interpretation, but they suffer from the exploding vs. vanishing gradients problem. Recent attempts to retain this simplicity while alleviating the gradient problem are based on proper initialization schemes or orthogonality/unitary constraints on the RNN's recurrence matrix, which, however, comes with limitations to its expressive power with regards to dynamical systems phenomena like chaos or multi-stability. Here, we instead suggest a regularization scheme that pushes part of the RNN's latent subspace toward a line attractor configuration that enables long short-term memory and arbitrarily slow time scales. We show that our approach excels on a number of benchmarks like the sequential MNIST or multiplication problems, and enables reconstruction of dynamical systems which harbor widely different time scales.

## 1 INTRODUCTION

Theories of complex systems in biology and physics are often formulated in terms of sets of stochastic differential or difference equations, i.e. as stochastic dynamical systems (DS). A long-standing desire is to retrieve these generating dynamical equations directly from observed time series data (Kantz & Schreiber, 2004). A variety of machine and deep learning methodologies toward this goal have been introduced in recent years (Chen et al., 2017; Champion et al., 2019; Jordan et al., 2019; Duncker et al., 2019; Ayed et al., 2019; Durstewitz, 2017; Koppe et al., 2019), many of them based on recurrent neural networks (RNN) which can universally approximate any DS (i.e., its flow field) under some mild conditions (Funahashi & Nakamura, 1993; Kimura & Nakano, 1998). However, vanilla RNN as often used in this context are well known for their problems in capturing long-term dependencies and slow time scales in the data (Hochreiter & Schmidhuber, 1997; Bengio et al., 1994). In DS terms, this is generally due to the fact that flexible information maintenance over long periods requires precise fine-tuning of model parameters toward 'line attractor' configurations (Fig. 1), a concept first propagated in computational neuroscience for addressing animal performance in parametric working memory tasks (Seung, 1996; Seung et al., 2000; Durstewitz, 2003). Line attractors introduce directions of zero-flow into the model's state space that enable long-term maintenance of arbitrary values (Fig. 1). Specially designed RNN architectures equipped with gating mechanisms and (linear) memory cells have been suggested for solving this issue (Hochreiter & Schmidhuber, 1997; Cho et al., 2014). However, from a DS perspective, simpler models that can more easily be analyzed and interpreted in DS terms, and for which more efficient inference algorithms exist that emphasize approximation of the true underlying DS would be preferable.

Recent solutions to the vanishing vs. exploding gradient problem attempt to retain the simplicity of vanilla RNN by initializing or constraining the recurrent weight matrix to be the identity (Le et al., 2015), orthogonal (Henaff et al., 2016; Helfrich et al., 2018) or unitary (Arjovsky et al., 2016). In this way, in a system including piecewise linear (PL) components like rectified-linear units (ReLU), line attractor dimensions are established from the start by construction or ensured throughout training by a specifically parameterized matrix decomposition. However, for many DS problems, line attractors instantiated by mere initialization procedures may be unstable and quickly dissolve during training. On the other hand, orthogonal or unitary constraints are too restrictive for reconstructing DS, and more generally from a computational perspective as well (Kerg et al., 2019): For instance, neither

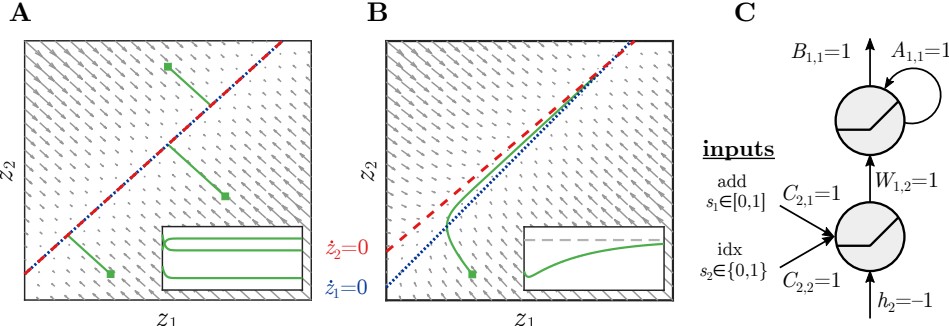

Fig. 1: Line attractors for solving long-time-scale problems. **A)**–**B**): Illustration of the state space of a 2-unit RNN (converted into a continuous time ODE, see Suppl. 7.1.2) with flow field (grey) and nullclines (set of points at which the flow of one of the variables vanishes, in blue and red). Insets: Time graphs of $z_1$ for $T = 30\,000$. **A**) Perfect line attractor. The flow converges to the line attractor from all directions and is exactly zero on the line, thus retaining states indefinitely in the absence of perturbations, as illustrated for 3 example trajectories (green) started from different initial conditions. **B**) Slightly detuned line attractor (cf. Durstewitz (2003)). The system's state still converges toward the 'line attractor ghost' (Strogatz, 2015), but then very slowly crawls up within the 'attractor tunnel' (green trajectory) until it hits the stable fixed point at the intersection of nullclines. Within the tunnel, flow velocity is smoothly regulated by the gap between nullclines, thus enabling arbitrary time constants. Note that along other, not illustrated dimensions of the system's state space the flow may still evolve freely in all directions. **C**) Simple 2-unit solution to the addition problem exploiting the line attractor properties of ReLUs in the positive quadrant. The output unit serves as a perfect integrator, while the input unit will only convey those input values to the output unit that are accompanied by a '1' in the second input stream (see 7.1.1 for complete parameters).

chaotic behavior (that requires diverging directions) nor settings with multiple isolated fixed point or limit cycle attractors are possible.

Here we therefore suggest a different solution to the problem, by pushing (but not strictly enforcing) ReLU-based, piecewise-linear RNN (PLRNN) toward line attractor configurations along some (but not all) directions in state space. We achieve this by adding special regularization terms for a subset of RNN units to the loss function that promote such a configuration. We demonstrate that our approach outperforms, or is en par with, LSTM and other, initialization-based, methods on a number of 'classical' machine learning benchmarks (Hochreiter & Schmidhuber, 1997). More importantly, we demonstrate that while with previous methods it was difficult to capture slow behavior in a DS that exhibits widely different time scales, our new regularization-supported inference efficiently captures all relevant time scales.

## 2  RELATED WORK

*Long-range dependency problems in RNN*. Error gradients in vanilla RNN trained by some form of gradient descent, like back-propagation through time (BPTT, Rumelhart et al. (1986)), tend to either explode or vanish due to the large product of derivative terms that results from recursive application of the chain rule over time steps (Hochreiter, 1991; Bengio et al., 1994; Hochreiter & Schmidhuber, 1997). Formally, RNN $z_t = F_\theta (z_{t-1}, s_t)$ are discrete time dynamical systems that tend to either converge, e.g. to fixed point or limit cycle attractors, or diverge (to infinity or as in chaotic systems) over time, unless parameters of the system are precisely tuned to create directions of zero-flow in the system's state space (Fig. 1), called line attractors (Seung, 1996; Seung et al., 2000; Durstewitz, 2003). Convergence of the RNN in general implies vanishing and global divergence exploding gradients. To address this issue, RNN with gated memory cells have been specifically designed (Hochreiter & Schmidhuber, 1997; Cho et al., 2014), but these are complicated and tedious to analyze from a DS perspective. Recently, Le et al. (2015) observed that initialization of the recurrent weight matrix $W$ to the identity in ReLU-based RNN may yield performance en par with LSTMs on standard machine learning benchmarks. For a ReLU with activity $z_t \geq 0$, zero bias and unit slope, this results in the identity mapping, hence a line attractor configuration. Talathi

& Vartak (2016) expanded on this idea by initializing the recurrence matrix such that its largest absolute eigenvalue is 1, arguing that this would leave other directions in the system's state space free for computations other than memory maintenance. Later work enforced orthogonal (Henaff et al., 2016; Helfrich et al., 2018; Jing et al., 2019) or unitary (Arjovsky et al., 2016) constraints on the recurrent weight matrix during training. While this appears to yield long-term memory performance superior to that of LSTMs, these networks are limited in their computational power (Kerg et al., 2019). This may be a consequence of the fact that RNN with orthogonal recurrence matrix are quite restricted in the range of dynamical phenomena they can produce, e.g. chaotic attractors are not possible since diverging eigen-directions are disabled. Our approach therefore is to establish line attractors only along some but not all directions in state space, and to only push the RNN toward these configurations but not strictly enforce them, such that convergence or divergence of RNN dynamics is still possible. We furthermore implement these concepts through regularization terms in the loss functions, such that they are encouraged throughout training unlike when only established through initialization.

*Dynamical systems reconstruction.* From a natural science perspective, the goal of reconstructing the underlying DS fundamentally differs from building a system that 'merely' yields good ahead predictions, as in DS reconstruction we require that the inferred model can freely reproduce (when no longer guided by the data) the underlying attractor geometries and state space properties (see section 3.5, Fig. S2; Kantz & Schreiber (2004)). Earlier work using RNN for DS identification (Roweis & Ghahramani, 2002; Yu et al., 2006) mainly focused on inferring the posterior over latent trajectories $\boldsymbol{Z} = \{\boldsymbol{z}_1, \ldots, \boldsymbol{z}_T\}$ given time series data $\boldsymbol{X} = \{\boldsymbol{x}_1, \ldots, \boldsymbol{x}_T\}$, $p(\boldsymbol{Z}|\boldsymbol{X})$, and on ahead predictions (Lu et al., 2017), hence did not show that inferred models can generate the underlying attractor geometries on their own. Others (Trischler & D'Eleuterio, 2016; Brunton et al., 2016) attempt to approximate the flow field, obtained e.g. by numerical differentiation, directly through basis expansions or neural networks, but numerical derivatives are problematic for their high variance and other numerical issues (Raissi, 2018; Baydin et al., 2018; Chen et al., 2017). Some approaches assume the form of the DS equations basically to be given (Raissi, 2018; Gorbach et al., 2017) and focus on estimating the system's latent states and parameters, rather than approximating an unknown DS based on the observed time series information alone. In many biological systems like the brain the intrinsic dynamics are highly stochastic with many noise sources, like probabilistic synaptic release (Stevens, 2003), such that models that do not explicitly account for dynamical process noise (Champion et al., 2019; Rudy et al., 2019) may be less suitable. Finally, some fully probabilistic models for DS reconstruction based on GRU (Fraccaro et al. (2016), cf. Jordan et al. (2019)), LSTM (Zheng et al., 2017), or radial basis function (Zhao & Park, 2017) networks are not easily interpretable and amenable to DS analysis. Most importantly, none of these previous approaches considers the long-range dependency problem within more easily tractable RNN for DS reconstruction.

## 3 MODEL FORMULATION AND OPTIMIZATION APPROACHES

### 3.1 MODEL AND PRELIMINARIES

Assume we are given two multivariate time series $\boldsymbol{S} = \{\boldsymbol{s}_t\}$ and $\boldsymbol{X} = \{\boldsymbol{x}_t\}$, one we will denote as 'inputs' ($\boldsymbol{S}$) and the other as 'outputs' ($\boldsymbol{X}$). We will first consider the 'classical' (supervised) machine learning setting where we wish to map $\boldsymbol{S}$ on $\boldsymbol{X}$ through a RNN with latent state equation $\boldsymbol{z}_t = F_\theta(\boldsymbol{z}_{t-1}, \boldsymbol{s}_t)$, as for instance in the 'addition problem' (Hochreiter & Schmidhuber, 1997). In DS reconstruction, in contrast, we usually have a dense time series $\boldsymbol{X}$ from which we wish to infer (unsupervised) the underlying DS, where $\boldsymbol{S}$ may provide an additional forcing function or sparse experimental inputs or perturbations.

The latent RNN we consider here takes the specific form

$$\boldsymbol{z}_t = \boldsymbol{A}\boldsymbol{z}_{t-1} + \boldsymbol{W}\phi(\boldsymbol{z}_{t-1}) + \boldsymbol{C}\boldsymbol{s}_t + \boldsymbol{h} + \boldsymbol{\varepsilon}_t, \ \ \boldsymbol{\varepsilon}_t \sim \mathcal{N}(0, \boldsymbol{\Sigma}), \tag{1}$$

where $\boldsymbol{z}_t \in \mathbb{R}^{M \times 1}$ is the hidden state (column) vector of dimension $M$, $\boldsymbol{A} \in \mathbb{R}^{M \times M}$ a diagonal and $\boldsymbol{W} \in \mathbb{R}^{M \times M}$ an off-diagonal matrix, $\boldsymbol{s}_t \in \mathbb{R}^{K \times 1}$ the external input of dimension $K$, $\boldsymbol{C} \in \mathbb{R}^{M \times K}$ the input mapping, $\boldsymbol{h} \in \mathbb{R}^{M \times 1}$ a bias, and $\boldsymbol{\varepsilon}_t$ a Gaussian noise term with diagonal covariance matrix $\text{diag}(\boldsymbol{\Sigma}) \in \mathbb{R}_+^M$. The nonlinearity $\phi(\boldsymbol{z})$ is a ReLU, $\phi(\boldsymbol{z})_i = \max(0, z_i), i \in \{1, \ldots, M\}$. This specific formulation is originally motivated by firing rate (population) models in computational neuroscience (Song et al., 2016; Durstewitz, 2017), where latent states $\boldsymbol{z}_t$ may represent

membrane voltages or currents, $\boldsymbol{A}$ the neurons' passive time constants, $\boldsymbol{W}$ the synaptic coupling among neurons, and $\phi(\cdot)$ the voltage-to-rate transfer function. However, for a RNN in the form $\boldsymbol{z}_t = \boldsymbol{W}\phi(\boldsymbol{z}_{t-1}) + \boldsymbol{h}$, note that the simple change of variables $\boldsymbol{y}_t \rightarrow \boldsymbol{W}^{-1}(\boldsymbol{z}_t - \boldsymbol{h})$ will yield the more familiar form $\boldsymbol{y}_t = \phi(\boldsymbol{W}\boldsymbol{y}_{t-1} + \boldsymbol{h})$ (Beer, 2006).

Besides its neuroscience motivation, note that by letting $\boldsymbol{A} = \boldsymbol{I}$, $\boldsymbol{W} = \boldsymbol{0}$, $\boldsymbol{h} = \boldsymbol{0}$, we get a strict line attractor system across the variables' whole support which we conjecture will be of advantage for establishing long short-term memory properties. Also we can solve for all of the system's fixed points analytically by solving the equations $\boldsymbol{z}^* = (\boldsymbol{I} - \boldsymbol{A} - \boldsymbol{W}\boldsymbol{D}_\Omega)^{-1}\boldsymbol{h}$, with $\boldsymbol{D}_\Omega$ as defined in Suppl. 7.1.2, and can determine their stability from the eigenvalues of matrix $\boldsymbol{A} + \boldsymbol{W}\boldsymbol{D}_\Omega$. We could do the same for limit cycles, in principle, which are fixed points of the $r$-times iterated map $F_{\boldsymbol{\theta}}^r$, although practically the number of configurations to consider increases exponentially as $2^{M \cdot r}$. Finally, we remark that a discrete piecewise-linear system can, under certain conditions, be transformed into an equivalent continuous-time (ODE) piecewise-linear system $\dot{\boldsymbol{\zeta}} = G_\Omega(\boldsymbol{\zeta}(t), \boldsymbol{s}(t))$ (Suppl. 7.1.2, Ozaki (2012)), in the sense that if $\boldsymbol{\zeta}(t) = \boldsymbol{z}_t$, then $\boldsymbol{\zeta}(t + \Delta t) = \boldsymbol{z}_{t+1}$ after a defined time step $\Delta t$. These are among the properties that make PLRNNs more amenable to rigorous DS analysis than other RNN formulations.

We will assume that the latent RNN states $\boldsymbol{z}_t$ are coupled to the actual observations $\boldsymbol{x}_t$ through a simple observation model of the form

$$\boldsymbol{x}_t = \boldsymbol{B}g(\boldsymbol{z}_t) + \boldsymbol{\eta}_t, \ \boldsymbol{\eta}_t \sim \mathcal{N}(0, \boldsymbol{\Gamma}) \tag{2}$$

in the case of real-valued observations $\boldsymbol{x}_t \in \mathbb{R}^{N \times 1}$, where $\boldsymbol{B} \in \mathbb{R}^{N \times M}$ is a factor loading matrix and $\mathrm{diag}(\boldsymbol{\Gamma}) \in \mathbb{R}_+^N$ the diagonal covariance matrix of the Gaussian observation noise, or

$$\hat{p}_{i,t} := \hat{p}_t(x_{i,t} = 1) = \left(e^{\boldsymbol{B}_{i,:}\boldsymbol{z}_t}\right) \left(\sum_{j=1}^N e^{\boldsymbol{B}_{j,:}\boldsymbol{z}_t}\right)^{-1}, \tag{3}$$

in the case of multi-categorical observations $x_{i,t} \in \{0, 1\}$, $\sum_i x_{i,t} = 1$.

## 3.2 REGULARIZATION APPROACH

We start from a similar idea as Le et al. (2015), who initialized RNN parameters such that it performs an identity mapping for $z_{i,t} \geq 0$. However, 1) we use a neuroscientifically motivated network architecture (eq. 1) that enables the identity mapping across the variables whole support, $z_{i,t} \in [-\infty, +\infty]$, 2) we encourage this mapping only for a subset $M_{\mathrm{reg}} \leq M$ of units (Fig. S1), leaving others free to perform arbitrary computations, and 3) we stabilize this configuration throughout training by introducing a specific $L_2$ regularization for parameters $\boldsymbol{A}$, $\boldsymbol{W}$, and $\boldsymbol{h}$ in eq. 1.

That way, we divide the units into two types, where the regularized units serve as a memory that tends to decay very slowly (depending on the size of the regularization term), while the remaining units maintain the flexibility to approximate any underlying DS, yet retaining the simplicity of the original RNN model (eq. 1). Specifically, the following penalty is added to the loss function (Fig. S1):

$$\mathrm{L}_{\mathrm{reg}} = \tau_A \sum_{i=1}^{M_{\mathrm{reg}}} (A_{i,i} - 1)^2 + \tau_W \sum_{i=1}^{M_{\mathrm{reg}}} \sum_{\substack{j=1 \\ j \neq i}}^{M} W_{i,j}^2 + \tau_h \sum_{i=1}^{M_{\mathrm{reg}}} h_i^2 \tag{4}$$

While this formulation allows us to trade off, for instance, the tendency toward a line attractor ($\boldsymbol{A} \rightarrow \boldsymbol{I}$, $\boldsymbol{h} \rightarrow \boldsymbol{0}$) vs. the sensitivity to other units' inputs ($\boldsymbol{W} \rightarrow \boldsymbol{0}$), for all experiments performed here a common value, $\tau_A = \tau_W = \tau_h = \tau$, was assumed for the three regularization factors.

## 3.3 OPTIMIZATION PROCEDURE FOR MACHINE LEARNING BENCHMARKS

For comparability with other approaches like LSTMs (Hochreiter & Schmidhuber, 1997) or iRNN (Le et al., 2015), we will assume that the latent state dynamics eq. 1 are deterministic (i.e., $\boldsymbol{\Sigma} = \boldsymbol{0}$), will take $g(\boldsymbol{z}_t) = \boldsymbol{z}_t$ and $\boldsymbol{\Gamma} = \boldsymbol{I}_N$ in eq. 2 (leading to an implicit Gaussian assumption with identity covariance matrix), and will use stochastic gradient descent (SGD) for training

to minimize the squared-error loss across $R$ samples, $\mathcal{L} = \sum_{n=1}^{R} \left( \hat{\boldsymbol{x}}_T^{(n)} - \boldsymbol{x}_T^{(n)} \right)^2$, between estimated and actual outputs for the addition and multiplication problems, and the cross entropy loss $\mathcal{L} = \sum_{n=1}^{R} \left( - \sum_{i=1}^{10} x_{i,T}^{(n)} \log(\hat{p}_{i,T}^{(n)}) \right)$ for sequential MNIST, to which penalty eq. 4 was added for the regularized PLRNN (rPLRNN). We used the Adam algorithm (Kingma & Ba, 2014) from the PyTorch package (Paszke et al., 2017) with a learning rate of 0.001, a gradient clip parameter of 10, and batch size of 16. In all cases, SGD is stopped after 100 epochs and the fit with the lowest loss across all epochs is chosen.

## 3.4 OPTIMIZATION PROCEDURE FOR DYNAMICAL SYSTEMS RECONSTRUCTION

For DS reconstruction we request that the latent RNN approximates the true generating system of equations, which is a taller order than learning the mapping $\boldsymbol{S} \to \boldsymbol{X}$ or predicting future values in a time series (cf. sect. 3.5). This point has important implications for the design of models, inference algorithms and performance metrics if the primary goal is DS reconstruction rather than 'mere' time series forecasting. In this context we consider the fully probabilistic, generative RNN eq. 1.

Together with eq. 2 (where we take $g(\boldsymbol{z}_t) = \phi(\boldsymbol{z}_t)$) this gives the typical form of a nonlinear state space model (Durbin & Koopman, 2012) with observation and process noise. We solve for the parameters $\boldsymbol{\theta} = \{\boldsymbol{A}, \boldsymbol{W}, \boldsymbol{C}, \boldsymbol{h}, \boldsymbol{\Sigma}, \boldsymbol{B}, \boldsymbol{\Gamma}\}$ by maximum likelihood, for which an efficient Expectation-Maximization (EM) algorithm has recently been suggested (Durstewitz, 2017; Koppe et al., 2019), which we will briefly summarize here. Since the involved integrals are not tractable, we start off from the evidence-lower bound (ELBO) to the log-likelihood which can be rewritten in various useful ways:

$$\log p(\boldsymbol{X}|\boldsymbol{\theta}) \geq \mathbb{E}_{\boldsymbol{Z} \sim q}[\log p_{\boldsymbol{\theta}}(\boldsymbol{X}, \boldsymbol{Z})] + H\left(q(\boldsymbol{Z}|\boldsymbol{X})\right)$$
$$= \log p(\boldsymbol{X}|\boldsymbol{\theta}) - D_{\mathrm{KL}}\left(q(\boldsymbol{Z}|\boldsymbol{X})\|p_{\boldsymbol{\theta}}(\boldsymbol{Z}|\boldsymbol{X})\right) =: \mathcal{L}\left(\boldsymbol{\theta}, q\right) \quad (5)$$

In the E-step, given a current estimate $\boldsymbol{\theta}^*$ for the parameters, we seek to determine the posterior $p_{\boldsymbol{\theta}}(\boldsymbol{Z}|\boldsymbol{X})$ which we approximate by a global Gaussian $q(\boldsymbol{Z}|\boldsymbol{X})$ instantiated by the maximizer (mode) $\boldsymbol{Z}^*$ of $p_{\boldsymbol{\theta}}(\boldsymbol{Z}|\boldsymbol{X})$ as an estimator of the mean, and the negative inverse Hessian around this maximizer as an estimator of the state covariance, i.e.

$$\mathbb{E}[\boldsymbol{Z}|\boldsymbol{X}] \approx \boldsymbol{Z}^* = \arg\max_{\boldsymbol{Z}} \log p_{\boldsymbol{\theta}}(\boldsymbol{Z}|\boldsymbol{X}) = \arg\max_{\boldsymbol{Z}} \left[\log p_{\boldsymbol{\theta}}(\boldsymbol{X}|\boldsymbol{Z}) + \log p_{\boldsymbol{\theta}}(\boldsymbol{Z}) - \log p_{\boldsymbol{\theta}}(\boldsymbol{X})\right]$$
$$= \arg\max_{\boldsymbol{Z}} \left[\log p_{\boldsymbol{\theta}}(\boldsymbol{X}|\boldsymbol{Z}) + \log p_{\boldsymbol{\theta}}(\boldsymbol{Z})\right], \quad (6)$$

since $\boldsymbol{Z}$ integrates out in $p_{\boldsymbol{\theta}}(\boldsymbol{X})$ (equivalently, this result can be derived from a Laplace approximation to the log-likelihood, $\log p(\boldsymbol{X}|\boldsymbol{\theta}) \approx \log p_{\boldsymbol{\theta}}(\boldsymbol{X}|\boldsymbol{Z}^*) + \log p_{\boldsymbol{\theta}}(\boldsymbol{Z}^*) - \frac{1}{2}\log|-\boldsymbol{L}^*| + \mathrm{const}$, where $\boldsymbol{L}^*$ is the Hessian evaluated at the maximizer). We solve this optimization problem by a fixed-point iteration scheme that efficiently exploits the model's piecewise linear structure (see Suppl. 7.1.3, Durstewitz (2017); Koppe et al. (2019)).

Using this approximate posterior for $p_{\boldsymbol{\theta}}(\boldsymbol{Z}|\boldsymbol{X})$, based on the model's piecewise-linear structure most of the expectation values $\mathbb{E}_{\boldsymbol{z} \sim q}[\phi(\boldsymbol{z})]$, $\mathbb{E}_{\boldsymbol{z} \sim q}[\phi(\boldsymbol{z})\boldsymbol{z}^\intercal]$, and $\mathbb{E}_{\boldsymbol{z} \sim q}[\phi(\boldsymbol{z})\phi(\boldsymbol{z})^\intercal]$, could be solved for (semi-)analytically (where $\boldsymbol{z}$ is the concatenated vector form of $\boldsymbol{Z}$, as in Suppl. 7.1.3). In the M-step, we seek $\boldsymbol{\theta}^* := \arg\max_{\boldsymbol{\theta}} \mathcal{L}(\boldsymbol{\theta}, q^*)$, assuming proposal density $q^*$ to be given from the E-step, which for a Gaussian observation model amounts to a simple linear regression problem (see Suppl. eq. 23). To force the PLRNN to really capture the underlying DS in its governing equations, we use a previously suggested (Koppe et al. 2019) stepwise annealing protocol that gradually shifts the burden of fitting the observations $\boldsymbol{X}$ from the observation model eq. 2 to the latent RNN model eq. 1 during training, the idea of which is to establish a mapping from latent states $\boldsymbol{Z}$ to observations $\boldsymbol{X}$ first, fixing this, and then enforcing the temporal consistency constraints implied by eq. 1 while accounting for the actual observations.

## 3.5 PERFORMANCE MEASURES

*Measures of prediction error.* For the machine learning benchmarks we employed the same criteria as used for optimization (MSE or cross-entropy, sect. 3.3) as performance metrics, evaluated across left-out test sets. In addition, we report the relative frequency $P_{\mathrm{correct}}$ of correctly predicted trials

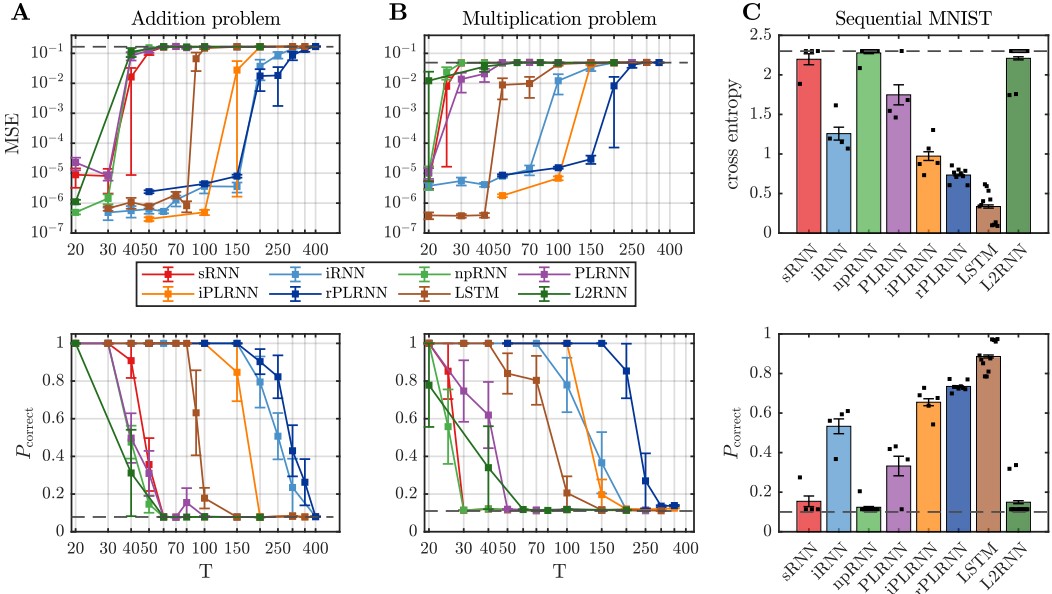

Fig. 2: Comparison of rPLRNN ($\tau = 5$, $\frac{M_{\text{reg}}}{M} = 0.5$, cf. Fig. S3) to other methods for **A**) addition problem, **B**) multiplication problem and **C**) sequential MNIST. Top row gives loss as a function of time series length $T$ (error bars = SEM), bottom row shows relative frequency of correct trials. Dashed lines indicate chance level, black dots in **C** indicate individual repetitions of the experiment.

across the test set. A correct trial in the addition and multiplication task is defined as an absolute prediction error smaller than 0.04 (analogous to Talathi & Vartak (2016)), while a correct trial in the sequential MNIST data set is defined as one for which the largest probability $\hat{p}_{i*} = \max_i \hat{p}_{i,T}$ indicated the correct class $x_{i*,T} = 1$.

*Agreement in attractor geometries.* From a DS perspective, it is not sufficient or even sensible to judge a method's ability to infer the underlying DS purely based on some form of (ahead-)prediction error like the MSE defined on the time series itself (Ch.12 in Kantz & Schreiber (2004)). Rather, we require that the inferred model can freely reproduce (when no longer guided by the data) the underlying attractor geometries and state space properties. This is not automatically guaranteed for a model that yields agreeable ahead predictions on a time series. Vice versa, if the underlying attractor is chaotic, with a tiny bit of noise even trajectories starting from the same initial condition will quickly diverge and ahead-prediction errors are not even meaningful as a performance metric (Fig. S2**A**). To quantify how well an inferred PLRNN captured the underlying dynamics we therefore followed Koppe et al. (2019) and used the Kullback-Leibler divergence between the true and reproduced probability distributions across states in state space, thus assessing the agreement in attractor geometries (cf. Takens (1981); Sauer et al. (1991)) rather than in precise matching of time series,

$$D_{\text{KL}} \left( p_{\text{true}}(\boldsymbol{x}) \| p_{\text{gen}}(\boldsymbol{x}|\boldsymbol{z}) \right) \approx \sum_{k=1}^{K} \hat{p}_{\text{true}}^{(k)}(\boldsymbol{x}) \log \left( \frac{\hat{p}_{\text{true}}^{(k)}(\boldsymbol{x})}{\hat{p}_{\text{gen}}^{(k)}(\boldsymbol{x}|\boldsymbol{z})} \right), \tag{7}$$

where $p_{\text{true}}(\boldsymbol{x})$ is the true distribution of observations across state space (not time!), $p_{\text{gen}}(\boldsymbol{x}|\boldsymbol{z})$ is the distribution of observations generated by running the inferred PLRNN, and the sum indicates a spatial discretization (binning) of the observed state space (see Suppl. 7.1.4 for more details). We emphasize that $\hat{p}_{\text{gen}}^{(k)}(\boldsymbol{x}|\boldsymbol{z})$ is obtained from freely *simulated* trajectories, i.e. drawn from the prior $\hat{p}(\boldsymbol{z})$, not from the inferred posteriors $\hat{p}(\boldsymbol{z}|\boldsymbol{x}_{\text{train}})$. (The form of $\hat{p}(\boldsymbol{z})$ is given by the dynamical model eq. 1 and has a 'mixture of piecewise-Gaussians' structure, see Koppe et al. (2019).) In addition, to assess reproduction of time scales by the inferred PLRNN, we computed the average correlation between the power spectra of the true and generated time series.

Table 1: Overview over the different models used for comparison

| NAME | DESCRIPTION |
|---|---|
| RNN | Vanilla ReLU based RNN |
| L2RNN | Vanilla ReLU RNN with standard $L_2$ regularization on all weights |
| iRNN | RNN with initialization $\boldsymbol{W}_0 = \boldsymbol{I}$ and $\boldsymbol{h}_0 = \boldsymbol{0}$ (Le et al., 2015) |
| npRNN | RNN with weights initialized to a normalized positive definite matrix with largest eigenvalue of 1 and biases initialized to zero (Talathi & Vartak, 2016) |
| PLRNN | PLRNN as given in eq. 1 (Koppe et al., 2019) |
| iPLRNN | PLRNN with initialization $\boldsymbol{A}_0 = \boldsymbol{I}$, $\boldsymbol{W}_0 = \boldsymbol{0}$ and $\boldsymbol{h}_0 = \boldsymbol{0}$ |
| rPLRNN | PLRNN initialized as illustrated in Fig. S1, with additional regularization term (eq. 4) during training |
| LSTM | Long Short-Term Memory (Hochreiter & Schmidhuber, 1997) |

## 4 NUMERICAL EXPERIMENTS

### 4.1 MACHINE LEARNING BENCHMARKS

We compared the performance of our rPLRNN to other models on the following three benchmarks requiring long short-term maintenance of information (as in Talathi & Vartak (2016) and Hochreiter & Schmidhuber (1997)): **1)** The *addition problem* of time length $T$ consists of $100\,000$ training and $10\,000$ test samples of $2 \times T$ input series $\boldsymbol{S} = \{\boldsymbol{s}_1, \ldots, \boldsymbol{s}_T\}$, where entries $\boldsymbol{s}_{1,:} \in [0,1]$ are drawn from a uniform random distribution and $\boldsymbol{s}_{2,:} \in \{0,1\}$ contains zeros except for two indicator bits placed randomly at times $t_1 < 10$ and $t_2 < T/2$. Constraints on $t_1$ and $t_2$ are chosen such that every trial requires a long memory of at least $T/2$ time steps. At the last time step $T$, the target output of the network is the sum of the two inputs in $\boldsymbol{s}_{1,:}$ indicated by the 1-entries in $\boldsymbol{s}_{2,:}$, $\boldsymbol{x}_T^{\text{target}} = s_{1,t_1} + s_{1,t_2}$. **2)** The *multiplication problem* is the same as the addition problem, only that the product instead of the sum has to be produced by the RNN as an output at time $T$, $\boldsymbol{x}_T^{\text{target}} = s_{1,t_1} \cdot s_{1,t_2}$. **3)** The MNIST dataset (LeCun & Cortes, 2010) consists of $60\,000$ training and $10\,000$ $28 \times 28$ test images of hand written digits. To make this a time series problem, in *sequential MNIST* the images are presented sequentially, pixel-by-pixel, scanning lines from upper left to bottom-right, resulting in time series of fixed length $T = 784$.

On all three benchmarks we compare the performance of the rPLRNN (eq. 1) to several other models summarized in Table 1. To achieve a meaningful comparison, all models have the same number of hidden states $M$, except for the LSTM, which requires three additional parameters for each hidden state and hence has only $M/4$ hidden states, yielding the overall same number of trainable parameters as for the other models. In all cases, $M = 40$, which initial numerical exploration suggested to be a good compromise between model complexity (bias) and data fit (variance) (Fig. S3).

Fig. 2 summarizes the results for the machine learning benchmarks. As can be seen, on the addition and multiplication tasks, and in terms of either the MSE or percentage correct, our rPLRNN outperforms all other tested methods, including LSTMs. Indeed, the LSTM performs even significantly worse than the iRNN and the iPLRNN. The large error bars in Fig. 2 result from the fact that the networks mostly learn these tasks in an all-or-none fashion, i.e. either learn the task and succeed in almost 100 percent of the cases or fail completely. The results for the sequential MNIST problem are summarized in Fig. 2**C**. While in this case the LSTM outperforms all other methods, the rPLRNN is almost en par with it. In addition, the iPLRNN outperforms the iRNN. Similar results were obtained for $M = 100$ units ($M = 25$, respectively, for LSTM; Fig. S6). While the rPLRNN in general outperformed the pure initialization-based models (iRNN, npRNN, iPLRNN), confirming that a line attractor subspace present at initialization may be lost throughout training, we conjecture that this difference in performance will become even more pronounced as noise levels or task complexity increase.

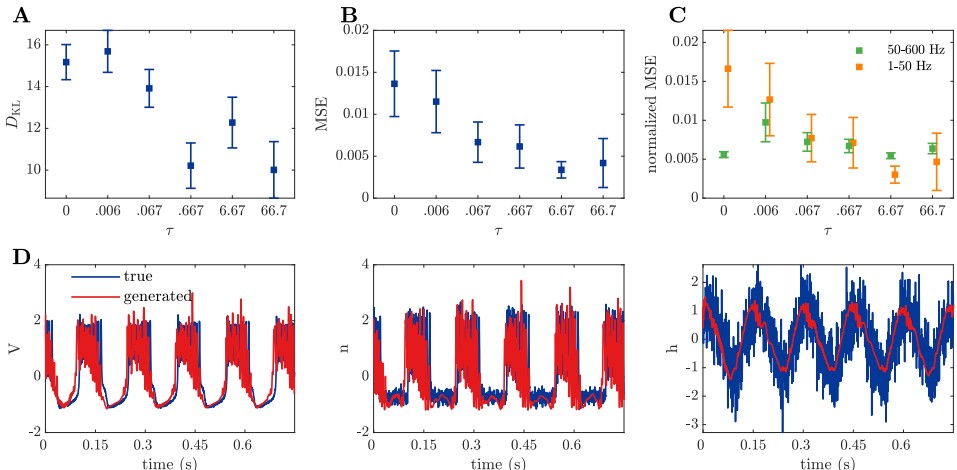

Fig. 3: Reconstruction of a 2-time scale DS (biophysical bursting neuron model) in limit cycle regime. **A**) KL divergence ($D_{KL}$) between true and generated state space distributions as a function of $\tau$. Unstable (globally diverging) system estimates were removed. **B**) Average MSE between power spectra of true and reconstructed DS. **C**) Average normalized MSE between power spectra of true and reconstructed DS split according to low ($\leq 50\,\text{Hz}$) and high ($> 50\,\text{Hz}$) frequency components. Error bars = SEM in all graphs. **D**) Example of (best) generated time series (red=reconstruction with $\tau = \frac{2}{3}$).

## 4.2 NUMERICAL EXPERIMENTS ON A DYNAMICAL SYSTEM WITH DIFFERENT TIME SCALES

Here our goal was to examine whether our regularization approach would also help with the identification of DS that harbor widely different time scales. By tuning systems in the vicinity of line attractors, multiple arbitrary time scales can be realized in theory (Durstewitz, 2003). To test this, we used a biophysically motivated (highly nonlinear) bursting cortical neuron model with one voltage and two conductance recovery variables (see Durstewitz (2009)), one slow and one fast (Suppl. 7.1.5). Reproduction of this DS is challenging since it produces very fast spikes on top of a slow nonlinear oscillation (Fig. 3**D**). Time series of standardized variables of length $T = 1500$ were generated from this model and provided as observations to the rPLRNN inference algorithm. rPLRNNs with $M = \{8 \dots 18\}$ states were estimated, with the regularization factor varied within $\tau \in \{0, 10^1, 10^2, 10^3, 10^4, 10^5\}/1500$.

Fig. 3**A** confirms our intuition that stronger regularization leads to better DS reconstruction as assessed by the KL divergence between true and generated state distributions. This decrease in $D_{KL}$ is accompanied by a likewise decrease in the MSE between the power spectra of true (Suppl. eq. 27) and generated (rPLRNN) voltage traces as $\tau$ increased (Fig. 3**B**). Fig. 3**D** gives an example of voltage traces and gating variables freely simulated (i.e., sampled) from the generative rPLRNN trained with $\tau = \frac{2}{3}$, illustrating that our model is in principle capable of capturing both the stiff spike dynamics and the slower oscillations in the second gating variable at the same time. Fig. 3**C** provides more insight into how the regularization worked: While the high frequency components ($> 50\,\text{Hz}$) related to the repetitive spiking activity hardly benefitted from increasing $\tau$, there was a strong reduction in the MSE computed on the power spectrum for the lower frequency range ($\leq 50\,\text{Hz}$), suggesting that increased regularization helps to map slowly evolving components of the dynamics.

## 5 CONCLUSIONS

In this work we have introduced a simple solution to the long short-term memory problem in RNN that on the one hand retains the simplicity and tractability of vanilla RNN, yet on the other hand does not curtail the universal computational capabilities of RNN (Koiran et al., 1994; Siegelmann & Sontag, 1995) and their ability to approximate arbitrary DS (Funahashi & Nakamura, 1993; Kimura & Nakano, 1998; Trischler & D'Eleuterio, 2016). We achieved this by adding regularization terms to the loss function that encourage the system to form a 'memory subspace', that is, line attractor

dimensions (Seung, 1996; Durstewitz, 2003) which would store arbitrary values for, if unperturbed, arbitrarily long periods. At the same time we did not rigorously enforce this constraint which has important implications for capturing slow time scales in the data: It allows the RNN to slightly depart from a perfect line attractor, which has been shown to constitute a general dynamical mechanism for regulating the speed of flow and thus the learning of arbitrary time constants that are not naturally included qua RNN design (Durstewitz, 2003; 2004). This is because as we come infinitesimally close to a line attractor and thus a bifurcation in the system's parameter space, the flow along this direction becomes arbitrarily slow until it vanishes completely in the line attractor configuration (Fig. 1). Moreover, part of the RNN's latent space was not regularized at all, leaving the system enough degrees of freedom for realizing arbitrary computations or dynamics. We showed that the rPLRNN is en par with or outperforms initialization-based approaches and LSTMs on a number of classical benchmarks, and, more importantly, that the regularization strongly facilitates the identification of challenging DS with widely different time scales in PLRNN-based algorithms for DS reconstruction. Future work will explore a wider range of DS models and empirical data with diverse temporal and dynamical phenomena. Another future direction may be to replace the EM algorithm by black-box variational inference, using the re-parameterization trick for gradient descent (Kingma & Welling, 2013; Rezende et al., 2014; Chung et al., 2015). While this would come with better scaling in $M$, the number of latent states (the scaling in $T$ is linear for EM as well, see Paninski et al. (2010)), the EM used here efficiently exploits the model's piecewise linear structure in finding the posterior over latent states and computing the parameters (see Suppl. 7.1.3). It may thus be more accurate and suitable for smaller-scale problems where high precision is required, as often encountered in neuroscience or physics.

## 6 ACKNOWLEDGEMENTS

This work was funded by grants from the German Research Foundation (DFG) to DD (Du 354/10-1, Du 354/8-2 within SPP 1665) and to GK (TRR265: A06 & B08). We would like to cordially thank Dr. Zahra Monfared for her careful reading of the manuscript and her thoughtful suggestions.

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

# 7 SUPPLEMENTARY MATERIAL

## 7.1 SUPPLEMENTARY TEXT

### 7.1.1 *Simple exact PLRNN solution for addition problem*

The exact PLRNN parameter settings (cf. eq. 1) for solving the addition problem with 2 units (cf. Fig. 1**C**) are as follows:

$$\boldsymbol{A} = \begin{pmatrix} 1 & 0 \\ 0 & 0 \end{pmatrix}, \boldsymbol{W} = \begin{pmatrix} 0 & 1 \\ 0 & 0 \end{pmatrix}, \boldsymbol{h} = \begin{pmatrix} 0 \\ -1 \end{pmatrix}, \boldsymbol{C} = \begin{pmatrix} 0 & 0 \\ 1 & 1 \end{pmatrix}, \boldsymbol{B} = \begin{pmatrix} 1 & 0 \end{pmatrix} \tag{8}$$

### 7.1.2 *Conversion from discrete to continuous time PLRNN*

Under some conditions we can translate the discrete into an equivalent continuous time PLRNN. Using $\boldsymbol{D}_{\Omega(t)}$ as defined below (7.1.3) for a single time step $t$, we can rewrite (ignoring the noise term and inputs) PLRNN eq. 1 in the form

$$\boldsymbol{z}_{t+1} = F(\boldsymbol{z}_t) = \boldsymbol{W}_{\Omega(t)} \boldsymbol{z}_t + \boldsymbol{h}, \text{ with } \boldsymbol{W}_{\Omega(t)} := \boldsymbol{A} + \boldsymbol{W} \boldsymbol{D}_{\Omega(t)}, \tag{9}$$

where $\Omega(t) := \{m | z_{m,t} > 0\}$ is the set of all unit indices with activation larger 0 at time $t$. To convert this into an equivalent (in the sense defined in eq. 11) system of (piecewise) ordinary differential equations (ODE), we need to find parameters $\widetilde{\boldsymbol{W}}_\Omega$ and $\widetilde{\boldsymbol{h}}$,

$$\dot{\boldsymbol{\zeta}} = G(\boldsymbol{\zeta}) = \widetilde{\boldsymbol{W}}_\Omega \boldsymbol{\zeta}(t) + \widetilde{\boldsymbol{h}}, \tag{10}$$

such that

$$\boldsymbol{z}_0 = \boldsymbol{\zeta}(0) \Rightarrow \boldsymbol{z}_1 = F(\boldsymbol{z}_0) = \boldsymbol{\zeta}(\Delta t), \tag{11}$$

where $\Delta t$ is the time step with which the empirically observed time series $\boldsymbol{X}$ was sampled. From these conditions it follows that for each of the $s \in \{1, \dots, 2^M\}$ subregions (orthants) defined by fixed index sets $\Omega^s \subseteq \{1, \dots, M\}$ we must have

$$(\boldsymbol{A} + \boldsymbol{W} \boldsymbol{D}_{\Omega^s} - \boldsymbol{I}) \boldsymbol{z}_0 + \boldsymbol{h} = \int_0^{\Delta t} \widetilde{\boldsymbol{W}}_{\Omega^s} \boldsymbol{\zeta}(t) + \widetilde{\boldsymbol{h}} \, dt, \tag{12}$$

where we assume that $\boldsymbol{D}_{\Omega^s}$ is constant for one time step, i.e. between 0 and $\Delta t$. We approach this by first solving the homogeneous system using the general ansatz for systems of linear ODEs,

$$(\boldsymbol{A} + \boldsymbol{W} \boldsymbol{D}_{\Omega^s} - \boldsymbol{I}) \boldsymbol{z}_0 \overset{!}{=} \int_0^{\Delta t} \widetilde{\boldsymbol{W}}_{\Omega^s} \sum_k c_k e^{\tilde{\lambda}_k t} \tilde{\boldsymbol{v}}_k \, dt \tag{13}$$

$$= \sum_k c_k \widetilde{\boldsymbol{W}}_{\Omega^s} \boldsymbol{v}_k \int_0^{\Delta t} e^{\tilde{\lambda}_k t} \, dt \tag{14}$$

$$= \sum_k c_k \tilde{\lambda}_k \boldsymbol{v}_k \frac{1}{\tilde{\lambda}_k} \left( e^{\tilde{\lambda}_k \Delta t} - 1 \right) \tag{15}$$

$$\Rightarrow \boldsymbol{W}_{\Omega^s} \boldsymbol{z}_0 \overset{!}{=} \sum_k c_k \boldsymbol{v}_k e^{\tilde{\lambda}_k \Delta t} \tag{16}$$

$$= \boldsymbol{V} \underbrace{\begin{pmatrix} e^{\tilde{\lambda}_1 \Delta t} & \cdots & \boldsymbol{0} \\ \vdots & \ddots & \vdots \\ \boldsymbol{0} & \cdots & e^{\tilde{\lambda}_M \Delta t} \end{pmatrix}}_{:=\boldsymbol{\Lambda}} \boldsymbol{c} \tag{17}$$

$$\Rightarrow \boldsymbol{W}_{\Omega^s} = \boldsymbol{V} \boldsymbol{\Lambda} \boldsymbol{V}^{-1}. \tag{18}$$

where we have used $\boldsymbol{z}_0 = \sum_k c_k \boldsymbol{v}_k$ on lines 15 and 16. Hence we can infer matrix $\widetilde{\boldsymbol{W}}_{\Omega^s}$ from the eigendecomposition of matrix $\boldsymbol{W}_{\Omega^s}$, by letting $\tilde{\lambda}_k = \frac{1}{\Delta t} \log \lambda_k$, where $\lambda_k$ are the eigenvalues of $\boldsymbol{W}_{\Omega^s}$, and reassembling

$$\widetilde{\boldsymbol{W}}_{\Omega^s} = \boldsymbol{V} \frac{1}{\Delta t} \log(\boldsymbol{\Lambda}) \boldsymbol{V}^{-1}. \tag{19}$$

We obtain the general solution for the inhomogeneous case by requiring that for all fixed points $\boldsymbol{z}^* = F(\boldsymbol{z}^*)$ of the map eq. 9 we have $G(\boldsymbol{z}^*) = 0$. Using this we obtain

$$\tilde{\boldsymbol{h}} = -\widetilde{\boldsymbol{W}}_{\Omega^s} \left(\boldsymbol{I} - \boldsymbol{W}_{\Omega^s}\right)^{-1} \boldsymbol{h} \tag{20}$$

Assuming inputs $\boldsymbol{s}_t$ to be constant across time step $\Delta t$, we can apply the same transformation to input matrix $\boldsymbol{C}$. Fig. S5 illustrates the discrete to continuous PLRNN conversion for a nonlinear oscillator.

Note that in the above derivations we have assumed that matrix $\boldsymbol{W}_{\Omega^s}$ can be diagonalized, and that all its eigenvalues are nonzero (in fact, $\boldsymbol{W}_{\Omega^s}$ should not have any negative real eigenvalues). In general, not every discrete-time PLRNN can be converted into a continuous-time ODE system in the sense defined above. For instance, we can have chaos in a 1d nonlinear map, while we need at least a 3d ODE system to create chaos (Strogatz, 2015).

### 7.1.3 *More details on EM algorithm*

Here we briefly outline the fixed-point-iteration algorithm for solving the maximization problem in eq. 6 (for more details see Durstewitz (2017); Koppe et al. (2019)). Given a Gaussian latent PLRNN and a Gaussian observation model, the joint density $p(\boldsymbol{X}, \boldsymbol{Z})$ will be piecewise Gaussian, hence eq. 6 piecewise quadratic in $\boldsymbol{Z}$. Let us concatenate all state variables across $m$ and $t$ into one long column vector $\boldsymbol{z} = (z_{1,1}, \ldots, z_{M,1}, \ldots, z_{1,T}, \ldots, z_{M,T})^{\mathsf{T}}$, arrange matrices $\boldsymbol{A}$, $\boldsymbol{W}$ into large $MT \times MT$ block tri-diagonal matrices, define $\boldsymbol{d}_\Omega \coloneqq \left(\mathbf{1}_{z_{1,1}>0}, \mathbf{1}_{z_{2,1}>0}, \ldots, \mathbf{1}_{z_{M,T}>0}\right)^{\mathsf{T}}$ as an indicator vector with a 1 for all states $z_{m,t} > 0$ and zeros otherwise, and $\boldsymbol{D}_\Omega \coloneqq \mathrm{diag}(\boldsymbol{d}_\Omega)$ as the diagonal matrix formed from this vector. Collecting all terms quadratic, linear, or constant in $\boldsymbol{z}$, we can then write down the optimization criterion in the form

$$Q_\Omega^*(\boldsymbol{z}) = -\frac{1}{2}[\boldsymbol{z}^{\mathsf{T}} \left(\boldsymbol{U}_0 + \boldsymbol{D}_\Omega \boldsymbol{U}_1 + \boldsymbol{U}_1^{\mathsf{T}} \boldsymbol{D}_\Omega + \boldsymbol{D}_\Omega \boldsymbol{U}_2 \boldsymbol{D}_\Omega\right) \boldsymbol{z} - \boldsymbol{z}^{\mathsf{T}} \left(\boldsymbol{v}_0 + \boldsymbol{D}_\Omega \boldsymbol{v}_1\right)$$
$$- \left(\boldsymbol{v}_0 + \boldsymbol{D}_\Omega \boldsymbol{v}_1\right)^{\mathsf{T}} \boldsymbol{z}] + \mathrm{const.} \tag{21}$$

In essence, the algorithm now iterates between the two steps:

1. Given fixed $\boldsymbol{D}_\Omega$, solve $\boldsymbol{z}^* = \left(\boldsymbol{U}_0 + \boldsymbol{D}_\Omega \boldsymbol{U}_1 + \boldsymbol{U}_1^{\mathsf{T}} \boldsymbol{D}_\Omega + \boldsymbol{D}_\Omega \boldsymbol{U}_2 \boldsymbol{D}_\Omega\right)^{-1} \left(\boldsymbol{v}_0 + \boldsymbol{D}_\Omega \boldsymbol{v}_1\right)$

2. Given fixed $\boldsymbol{z}^*$, recompute $\boldsymbol{D}_\Omega$

until either convergence or one of several stopping criteria (partly likelihood-based, partly to avoid loops) is reached. The solution may afterwards be refined by one quadratic programming step. Numerical experiments showed this algorithm to be very fast and efficient (Durstewitz, 2017; Koppe et al., 2019). At $\boldsymbol{z}^*$, an estimate of the state covariance is then obtained as the inverse negative Hessian,

$$\boldsymbol{V} = \left(\boldsymbol{U}_0 + \boldsymbol{D}_\Omega \boldsymbol{U}_1 + \boldsymbol{U}_1^{\mathsf{T}} \boldsymbol{D}_\Omega + \boldsymbol{D}_\Omega \boldsymbol{U}_2 \boldsymbol{D}_\Omega\right)^{-1}. \tag{22}$$

In the M-step, using the proposal density $q^*$ from the E-step, the solution to the maximization problem $\boldsymbol{\theta}^* \coloneqq \arg\max_{\boldsymbol{\theta}} \mathcal{L}(\boldsymbol{\theta}, q^*)$, can generally be expressed in the form

$$\boldsymbol{\theta}^* = \left(\sum_t \mathbb{E}\left[\boldsymbol{\alpha}_t \boldsymbol{\beta}_t^{\mathsf{T}}\right]\right) \left(\sum_t \mathbb{E}\left[\boldsymbol{\beta}_t \boldsymbol{\beta}_t^{\mathsf{T}}\right]\right)^{-1}, \tag{23}$$

where, for the latent model, eq. 1, $\boldsymbol{\alpha}_t = \boldsymbol{z}_t$ and $\boldsymbol{\beta}_t \coloneqq \left[\boldsymbol{z}_{t-1}^{\mathsf{T}}, \phi(\boldsymbol{z}_{t-1})^{\mathsf{T}}, \boldsymbol{s}_t^{\mathsf{T}}, 1\right]^{\mathsf{T}} \in \mathbb{R}^{2M+K+1}$, and for the observation model, eq. 2, $\boldsymbol{\alpha}_t = \boldsymbol{x}_t$ and $\boldsymbol{\beta}_t = g(\boldsymbol{z}_t)$.

### 7.1.4 *More details on DS performance measure*

The measure $D_{\mathrm{KL}}$ introduced in the main text for assessing the agreement in attractor geometries only works for situations where the ground truth $p_{\mathrm{true}}(\boldsymbol{X})$ is known. Following Koppe et al. (2019), here we would like to briefly indicate how a proxy for $D_{\mathrm{KL}}$ may be obtained in empirical situations where no ground truth is available. Reasoning that for a well reconstructed DS the inferred posterior $p_{\mathrm{inf}}(\boldsymbol{z}|\boldsymbol{x})$ given the observations should be a good representative of the prior generative

dynamics $p_{\text{gen}}(\boldsymbol{z})$, one may use the Kullback-Leibler divergence between the distribution over latent states, obtained by sampling from the prior density $p_{\text{gen}}(\boldsymbol{z})$, and the (data-constrained) posterior distribution $p_{\text{inf}}(\boldsymbol{z}|\boldsymbol{x})$ (where $\boldsymbol{z} \in \mathbb{R}^{M \times 1}$ and $\boldsymbol{x} \in \mathbb{R}^{N \times 1}$), taken across the system's state space:

$$D_{\text{KL}}\left(p_{\text{inf}}(\boldsymbol{z}|\boldsymbol{x})\|p_{\text{gen}}(\boldsymbol{z})\right) = \int_{\boldsymbol{z} \in \mathbb{R}^{M \times 1}} p_{\text{inf}}(\boldsymbol{z}|\boldsymbol{x}) \log \frac{p_{\text{inf}}(\boldsymbol{z}|\boldsymbol{x})}{p_{\text{gen}}(\boldsymbol{z})} d\boldsymbol{z} \tag{24}$$

As evaluating this integral is difficult, one could further approximate $p_{\text{inf}}(\boldsymbol{z}|\boldsymbol{x})$ and $p_{\text{gen}}(\boldsymbol{z})$ by Gaussian mixtures across trajectories, i.e. $p_{\text{inf}}(\boldsymbol{z}|\boldsymbol{x}) \approx \frac{1}{T}\sum_{t=1}^{T} p(\boldsymbol{z}_t|\boldsymbol{x}_{1:T})$ and $p_{\text{gen}}(\boldsymbol{z}) \approx \frac{1}{L}\sum_{l=1}^{L} p(\boldsymbol{z}_l|\boldsymbol{z}_{l-1})$, where the mean and covariance of $p(\boldsymbol{z}_t|\boldsymbol{x}_{1:T})$ and $p(\boldsymbol{z}_l|\boldsymbol{z}_{l-1})$ are obtained by marginalizing over the multivariate distributions $p(\boldsymbol{Z}|\boldsymbol{X})$ and $p_{\text{gen}}(\boldsymbol{Z})$, respectively, yielding $\mathbb{E}[\boldsymbol{z}_t|\boldsymbol{x}_{1:T}]$, $\mathbb{E}[\boldsymbol{z}_l|\boldsymbol{z}_{l-1}]$, and covariance matrices $\text{Var}(\boldsymbol{z}_t|\boldsymbol{x}_{1:T})$ and $\text{Var}(\boldsymbol{z}_l|\boldsymbol{z}_{l-1})$. Supplementary eq. 24 may then be numerically approximated through Monte Carlo sampling (Hershey & Olsen, 2007) by

$$D_{\text{KL}}\left(p_{\text{inf}}(\boldsymbol{z}|\boldsymbol{x})\|p_{\text{gen}}(\boldsymbol{z})\right) \approx \frac{1}{n}\sum_{i=1}^{n} \log \frac{p_{\text{inf}}(\boldsymbol{z}^{(i)}|\boldsymbol{x})}{p_{\text{gen}}(\boldsymbol{z}^{(i)})}, \quad \boldsymbol{z}^{(i)} \sim p_{\text{inf}}(\boldsymbol{z}|\boldsymbol{x}) \tag{25}$$

For high-dimensional state spaces, for which MC sampling becomes challenging, there is luckily a variational approximation of eq. 24 available (Hershey & Olsen, 2007):

$$D_{\text{KL}}^{\text{variational}}\left(p_{\text{inf}}(\boldsymbol{z}|\boldsymbol{x})\|p_{\text{gen}}(\boldsymbol{z})\right) \approx \frac{1}{T}\sum_{t=1}^{T} \log \frac{\sum_{j=1}^{T} e^{-D_{\text{KL}}(p(\boldsymbol{z}_t|\boldsymbol{x}_{1:T})\|p(\boldsymbol{z}_j|\boldsymbol{x}_{1:T}))}}{\sum_{k=1}^{T} e^{-D_{\text{KL}}(p(\boldsymbol{z}_t|\boldsymbol{x}_{1:T})\|p(\boldsymbol{z}_k|\boldsymbol{z}_{k-1}))}}, \tag{26}$$

where the KL divergences in the exponentials are among Gaussians for which we have an analytical expression.

### 7.1.5 *More details on single neuron model*

The neuron model used in section 4.2 is described by

$$-C_m \dot{V} = g_L(V - E_L) + g_{Na}m_\infty(V)(V - E_{Na}) + g_K n(V - E_K)$$
$$+ g_M h(V - E_K) + g_{NMDA}\sigma(V)(V - E_{NMDA}) \tag{27}$$

$$\dot{h} = \frac{h_\infty(V) - h}{\tau_h} \tag{28}$$

$$\dot{n} = \frac{n_\infty(V) - n}{\tau_n} \tag{29}$$

$$\sigma(V) = \left[1 + .33e^{-.0625V}\right]^{-1} \tag{30}$$

where $C_m$ refers to the neuron's membrane capacitance, the $g_\bullet$ to different membrane conductances, $E_\bullet$ to the respective reversal potentials, and $m$, $h$, and $n$ are gating variables with limiting values given by

$$\{m_\infty, n_\infty, h_\infty\} = \left[1 + e^{(\{V_{hNa}, V_{hK}, V_{hM}\} - V)/\{k_{Na}, k_K, k_M\}}\right]^{-1} \tag{31}$$

Different parameter settings in this model lead to different dynamical phenomena, including regular spiking, slow bursting or chaos (see Durstewitz (2009) for details). Parameter settings used here were: $C_m = 6\,\mu\text{F}$, $g_L = 8\,\text{mS}$, $E_L = -80\,\text{mV}$, $g_{Na} = 20\,\text{mS}$, $E_{Na} = 60\,\text{mV}$, $V_{hNa} = -20\,\text{mV}$, $k_{Na} = 15$, $g_K = 10\,\text{mS}$, $E_K = -90\,\text{mV}$, $V_{hK} = -25\,\text{mV}$, $k_K = 5$, $\tau_n = 1\,\text{ms}$, $g_M = 25\,\text{mS}$, $V_{hM} = -15\,\text{mV}$, $k_M = 5$, $\tau_h = 200\,\text{ms}$, $g_{NMDA} = 10.2\,\text{mS}$.

## 7.2 SUPPLEMENTARY FIGURES

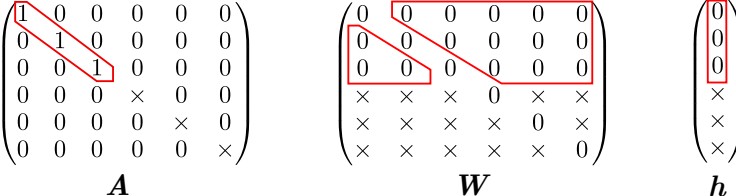

Fig. S1: Illustration of the L2-regularization for the PLRNN's auto-regression matrix $A$, coupling matrix $W$, and bias terms $h$. Regularized values are indicated in red, crosses mark arbitrary values (all other values set to 0 as indicated).

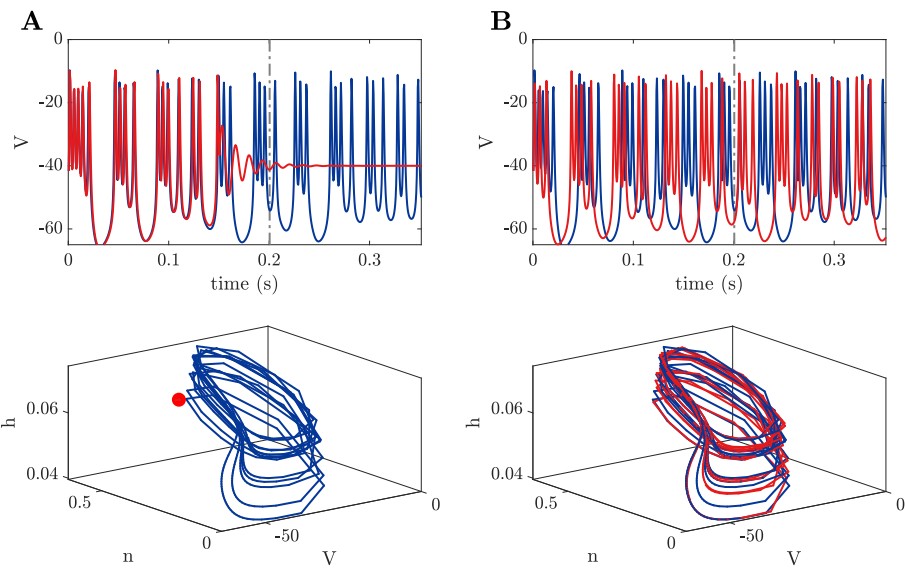

Fig. S2: MSE evaluated between time series is not a good measure for DS reconstruction. **A**) Time graph (top) and state space (bottom) for the single neuron model (see section 4.2 and Suppl. 7.1.5) with parameters in the chaotic regime (blue curves) and with simple fixed point dynamics in the limit (red line). Although the system has vastly different limiting behaviors (attractor geometries) in these two cases, as visualized in the state space, the agreement in time series initially seems to indicate a perfect fit. **B**) Same as in **A**) for two trajectories drawn from exactly the same DS (i.e., same parameters) with slightly different initial conditions. Despite identical dynamics, the trajectories immediately diverge, resulting in a high MSE. Dash-dotted grey lines in top graphs indicate the point from which onward the state space trajectories were depicted.

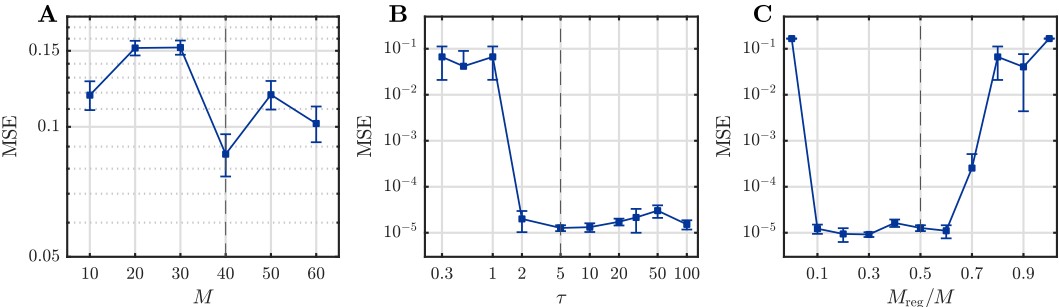

Fig. S3: Performance of the rPLRNN on the addition problem for different **A**) numbers of latent states $M$, **B**) values of $\tau$ and **C**) proportions $M_{\mathrm{reg}}/M$. Dashed lines denote the values used for the results reported in section 4.1

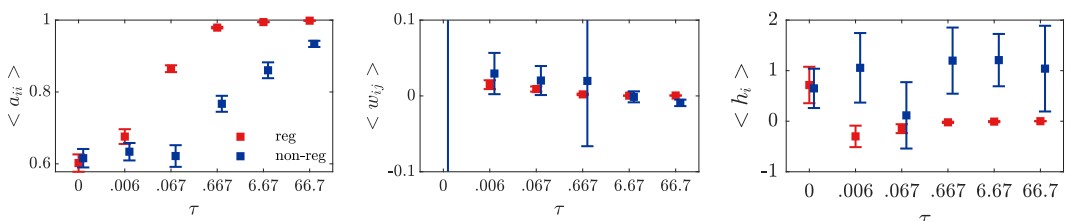

Fig. S4: Effect of regularization strength $\tau$ on rPLRNN network parameters (cf. eq. 1) (regularized parameters for states $m \leq M_{\mathrm{reg}}$, eq. 1, in red). Note that some of the non-regularized network parameters (in blue) appear to systematically change as well as $\tau$ is varied.

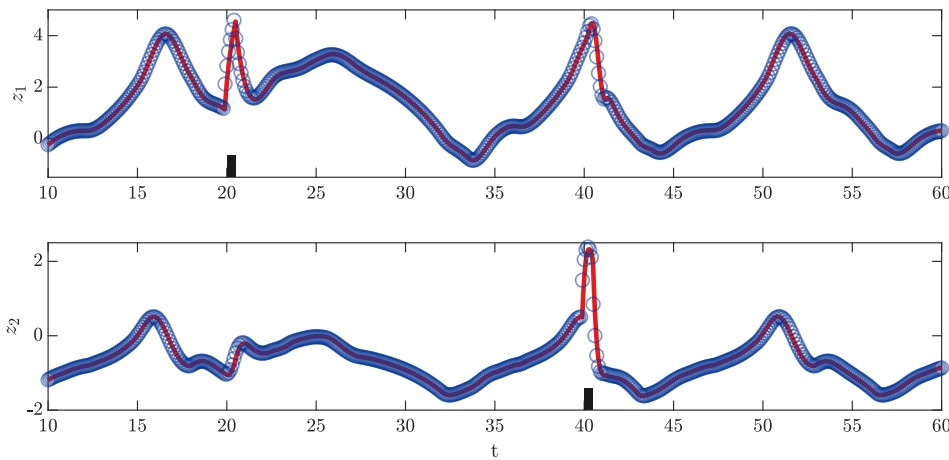

Fig. S5: Illustration of conversion of discrete into continuous time PLRNN for a PLRNN emulation of the nonlinear van-der-Pol oscillator. Shown are the first two latent dimensions. Red lines: continuous solution; blue circles: discrete solution; black bars: perturbations (external inputs).

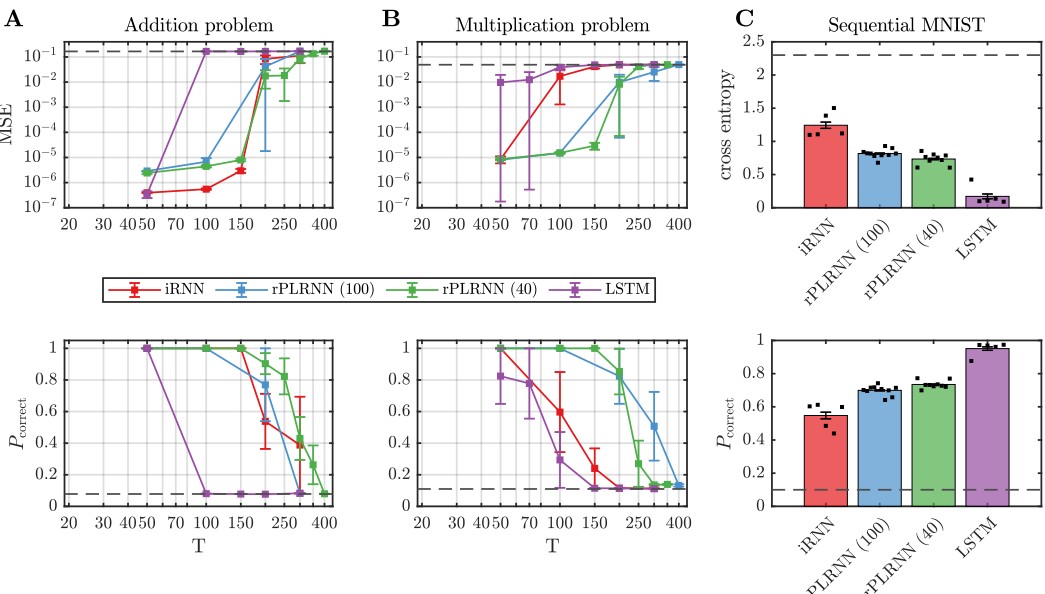

Fig. S6: Comparison of rPLRNN ($\tau = 5$, $\frac{M_{\text{reg}}}{M} = 0.5$, cf. Fig. S3) for $M = 40$ and $M = 100$ to iRNN ($M = 100$) and LSTM ($M = 100/4$) for **A**) addition problem, **B**) multiplication problem and **C**) sequential MNIST. Top row gives loss as a function of time series length $T$ (error bars = SEM), bottom row shows relative frequency of correct trials. Dashed lines indicate chance level, black dots in **C** indicate individual repetitions of the experiment. Note that the rPLRNN does not improve for $M = 100$ vs. $M = 40$.

