# OpenReview forum: "Inferring Dynamical Systems with Long-Range Dependencies through Line Attractor Regularization"
_ICLR.cc/2020/Conference — Reject_

### Official Review · AnonReviewer2 · 2019-10-19
**Official Blind Review #2**

**Rating:** 3

**Review:**

This paper presents a regulariser that encourages the formation of line attractors in RNNs. This regulariser works on a neuroscience-motivated formulation of RNN, bringing the Jacobian of the dynamic system close to identity. This paper is well-written, with a good coverage of background material from machine learning to computational neuroscience.  While the alleviating the exploding and vanishing gradient problem for simple RNNs is an interesting direction, I think the empirical results are not sufficient to support claims in the paper.

The paper starts by criticising initialisation and reparametrisation-based techniques. However, I am not convinced why such methods limit the expressive power of RNNs. In fact, one may argue that initialisation is a milder constraint compared with an explicit regulariser, since regularisation affects the entire learning process. It seems that such initialisation requires less tuning (i.e., just identity) compared with the regulariser weights (a rather wide range of choices). Either a theoretical justification or strong empirical results are required to support this claim. However, both are missing in the current paper.

Experiments:

First, the baseline results for even toy problems (e.g., addition) are unclear. Despite the results in the paper show the advantage of the proposed method, direct comparison with results from other papers are missing. For example,  when the T > 150 the results from Le et al, 2015 (A Simple Way to Initialize Recurrent Networks of
Rectified Linear Units) were much better compared with baseline results in the paper. This could be due to the smaller size of the models (40 vs 100 hidden units in Le et al.). For clear and direct comparison in this case, models with comparable size should be used in these experiments. Despite this, I wonder why the performance of addition and multiplication seem even worse than the much smaller model reported in Hochreiter and Schmidhuber 1997? (see table 7 and 8)?

Actually, it would be helpful to test the proposed method on more practical tasks such as language (at least synthetic ones) and speech modelling, which would bring more impact on the wider community.

A few technical questions:

- Is the form of eq. 1 necessary, or can the method be adapted for the more standard formulation of RNN used in machine learning? It seems that A can be interpreted as a skip connection

- Eq.3 can simply be referred to as “softmax”

- Please comment on the algorithm in section 3.4 in comparison with more standard variational approaches, such as stochastic variational inference with reparametrisation as in variational RNNs (e.g., Chung, et al., 2015, A Recurrent Latent Variable Model for Sequential Data). Is “stepwise annealling” always necessary?

- Eq. 7 as a measurement of the match between trajectories still depends on z. Is there an additional expectation over p(z)? Is “freely simulated trajectories” the prior over trajectories? If so, what’s the form of this prior?

**Experience Assessment:**

I have published one or two papers in this area.

**Review Assessment: Checking Correctness Of Derivations And Theory:**

I assessed the sensibility of the derivations and theory.

**Review Assessment: Checking Correctness Of Experiments:**

I carefully checked the experiments.

**Review Assessment: Thoroughness In Paper Reading:**

I read the paper at least twice and used my best judgement in assessing the paper.

---

> ### Author Response · Authors · 2019-11-06
> **point-by-point reply, theoretical justifaction & experiments**
>
> The limitations of orthogonal/unitary RNN are well illustrated in the recent work by Kerg et al. (https://arxiv.org/abs/1905.12080), but we also have a formal proof that orthogonal RNN cannot exhibit certain dynamical systems phenomena like, e.g., chaos (briefly, among other things, because they do not have diverging eigen-directions). We would be happy to add this proof to the manuscript if that is considered useful.
>
> Regarding initialization, in all our experiments all the initialization-based approaches always performed worse than the regularization, and we suspect this is the case since over the course of training the initial condition often simply gets lost (after all, the training [parameter updating] constitutes a stochastic dynamical systems process as well), and with it the ability to express long short-term memory. This certainly will depend on the complexity of the task, the length of the delays used, and the amount of noise/stochasticity in the process, however, where we’d expect that our approach gains more and more of an edge as T, noise, and task complexity increase (playing against pure initialization-based approaches).
>
> Re experiments, we will attempt to add runs with 100 units at least for iRNN and rPLRNN, and comparable # of parameters for LSTM, if we’ll manage in the time available for the revision!
> We emphasize, however, that we settled on 40 units since this appeared to optimize the bias-variance tradeoff in our case, and hence is a reasonable choice from a statistical learning perspective. Given this, we put all different models on equal footing by equalizing their degrees of freedom as much as possible.
> We are not quite sure why the results for LSTMs in H&S97 are better than ours, but will surely look into this – perhaps it’s related to the optimizer we used (Adam) or the specific learning protocol or …; we note that we will provide all our code (now available via the Dropbox link) publicly on github, so anyone is free to check our code and results for possible clues.
> We aimed to standardize and make comparable the protocols & conditions among all RNN models tested to the largest degree possible, however, so specific performance differences we get in comparison to the literature are likely to affect all models in our competition, including the one advocated by us.
>
> While language modeling would certainly be interesting, it’s probably not something we’ll achieve in the short time available for revision; we used the three benchmarks which we feel are the ones most widely used in the literature on the vanish./expl. grad. problem, with language modeling less commonly employed.
> Ultimately, however, our interest is much more in recovering underlying nonlinear dynamical systems in a fully generative sense (Fig. 3). In our minds we have achieved a major step here, but it appears this point unfortunately got somewhat lost perhaps by the focus on machine learning benchmarks in the first part.
>
> On the technical questions:
>
> Yes, the model could largely be rewritten in ‘standard machine learning form’ (see lines below eq. 1, by a change of variables). However, the specific separation of A and W we chose, at least in theory enables a line attractor to be formed across the whole support of the variables (the whole real line), and therefore should be beneficial for this purpose. Moreover, it makes an explicit connection to models used in computational neuroscience, a feature it appears that referee #3 liked.
>
> Softmax: Yes, true; we wanted to be explicit here, but are happy to change this and spare one standard equation.
>
> Comparison to var. RNN: Very fair point, please see our reply to referee #3 who brought up the same issue.
> In brief, we actually did implement the same model using a variational approach and the re-parameterization trick, but it performs (much) worse than the EM-based algorithm which is specifically designed for nonlinear dynamical systems reconstruction, and which also requires (much) less data. The annealing may not always be necessary, but at least for the purpose of reconstructing dynamical systems it improves performance by orders of magnitude (see Koppe et al. 2019 PLoS Comp. Biol.)!
> As already mentioned in our reply to referee #3, we will discuss pro’s and con’s of the different approaches in the revised manuscript as requested.
>
> Re eq. 7, measure of match between attractor geometries: Yes it depends on z, but the whole ‘measure’ is across state space, not time. And yes, by freely simulated we mean drawing samples from the generative model, i.e. the prior p(z), and using that to produce samples from p(x|z). The prior is determined by our specific model formulation, eq. 1, i.e. at each time step t it is conditionally (on z_{t-1}) Gaussian, but overall the prior across latent trajectories will be a high-dimensional mixture of piecewise Gaussians (owing to the piecewise linear, ReLU-based structure of the RNN).
> We will try to clarify all these points in Suppl. sect. 6.1.4.

---

### Official Review · AnonReviewer3 · 2019-10-23
**Official Blind Review #3**

**Rating:** 6

**Review:**

The paper proposes a regularization scheme to improve the ability of RNNs in capturing long-range dependency in the latent space. The proposed model then uses EM for inference and achieves superior performance in sequence modeling tasks over LSTMs and iRNNs.

+ The motivation of the line attractor is novel and effective. The special RNN model studied in the paper has strong connections with neuro-dynamics models.
+ The paper is well motivated and clearly written. The illustration about the line attractor is particularly interesting.
+ Good experimental performance on multiple sequence modeling tasks including addition, multiplication and sequence MNIST.

- The paper is building a generative model for sequences. It’s not clear to me why VAE or variational RNN type of approaches cannot be used in this setting. One might think of replacing the Gaussian prior with more complex distributions. The inference procedure can also be significantly simplified with variational inference.
- The step-wise annealing together with EM inference is not scalable, which prohibits the model from applying to large-scale sequence modeling tasks.

Minor comments
- " All code used in this work is freely available on the github site ... . " Remove this sentence


**Experience Assessment:**

I have published one or two papers in this area.

**Review Assessment: Checking Correctness Of Derivations And Theory:**

I carefully checked the derivations and theory.

**Review Assessment: Checking Correctness Of Experiments:**

I carefully checked the experiments.

**Review Assessment: Thoroughness In Paper Reading:**

I read the paper thoroughly.

---

> ### Author Response · Authors · 2019-11-06
> **VAE/ var. RNN vs. EM**
>
> At least one referee seems to somewhat like our approach, thank you!
>
> Re the ‘-‘ points:
>
> - Absolutely, VAE/var. RNN could be used as well, and in fact we have already implemented & tested this in our group. However, at this point EM produces much better results regarding reconstruction of the underlying dynamical system, possibly because in our algorithm & model we can compute expectations over latent states, as well as parameters given these expectations (M-step), exactly & analytically, and rely on efficient fixed-point iterations for maximizing the ELBO in the E-step (see sect. 6.1.3). Hence our algorithm is sort of optimized for this problem of reconstructing nonlinear dynamical systems, exploiting the piecewise linear formulation (PLRNN), and therefore is to be preferred for smaller-scale problems as often encountered in physics or neuroscience. In fact, we observed that it gets away with much less data than what we would need for our VAE implementation!
>
> - Yes, this is the downside of our algorithm (and also what ultimately motivated us to re-implement the whole approach in a VAE framework). We point out, however, that our algorithm is still linear in T (the length of the time series) since we can effectively exploit the block-tridiagonal structure of the Hessian for inversion (see also Paninski et al. 2010, J of Comput. Neurosci.). So for smaller-size problems as in physics or neuroscience experiments, we clearly recommend using EM for the reasons mentioned above (the more exact approach requiring less data), but for larger-scale problems one probably would need to switch to a variational framework.
>
> We are happy to include this discussion of up- & downsides of the different approaches in the revision.

---

### Official Review · AnonReviewer1 · 2019-10-24
**Official Blind Review #1**

**Rating:** 1

**Review:**

Overview
This paper proposes a type of regularization for recurrent networks, with the goal of encouraging particular dynamical structures (in this case, line attractors) in the dynamics of the networks. The proposed regularization penalty is only applied to a subset of the recurrent units; the motivation for this is to allow neurons not contained in the subset to learn different structures. The paper applies this regularization method on three example machine learning sequence tasks: an addition task, a multiplication task, and sequential MNIST classification, as well as on learning a 2-D dynamical system model of a bursting neuron with two different timescales.

Major comments
I have a number of serious concerns about the paper's motivation, logic, and experiments:

- First, the paper motivates the proposed regularization as a way to encourage the network to have line attractor dynamics. In particular, the paper dismisses gated architectures as not being interpretable, stating that LSTMs and GRUs "are complicated and tedious to analyze from a DS perspective." (pg 2). However, there is recent work both theoretical (https://arxiv.org/abs/1906.01005) and empirical (https://arxiv.org/abs/1907.08549) that analyzes these gated architectures from a DS perspective. In particular, these papers demonstrate that LSTMs and GRUs are perfectly capable of learning line attractors. Given that it is possible to analyze gated architectures as dynamical systems, the overall motivation of the paper is much weaker.

- The paper proposes a squared penalty on subsets of the weights in the recurrent network as a way to encourage line attractor dynamics. However, it is not clear to me that this is sufficient. In particular, unless the subset of the network that implements the line/plane attractor is completely disconnected from the rest of the network, then the overall dynamics may not contain a line attractor (the units will interact with the unregularized units). Also, the proposed regularization penalty only penalizes the diagonal elements of the A matrix to be close to 1--but shouldn't the off diagonal elements also be penalized to be close to zero?

- Moreover, the paper makes no mention of the Jacobian of these recurrent networks. The eigenvalues of the Jacobian of the recurrent networks determine the behavior of the linearized system around fixed points--specifically, eigenvalues with real part close to 1 will exhibit slow dynamics (approximate line attractors along those dimensions). It seems to me that a much more natural way of encouraging line attractor dynamics is to place a regularization penalty on the Jacobian itself (which is analytically tricky, but numerically more plausible with modern autodifferentiation software). Regardless, the authors should compare the eigenvalues of the recurrent networks' Jacobian when using their regularization method vs without it. Does the proposed regularization encourage the Jacobian of the resulting networks to have eigenvalues close to 1?

- The paper compares the proposed method with a number of vanilla RNNs with different initializations, and an LSTM. However, a critical missing baseline is simply an RNN with l2 regularization on the weights (standard regularization in the literature). This baseline is important to determine if the proposed regularization simply helps because it is an l2 penalty on the weights (note that none of the other baselines have regularization).

- The paper motivates the method as trying to study line attractor dynamics, but then does not apply them to tasks where line attractors are required. For example, the addition and multiplication tasks require discrete memories, not line attractors. The bursting neuron approximation (2D dynamical system) also does not involve a line attractor. However, there definitely exist tasks both in neuroscience (e.g. sensory integration in decision making, path integration in navigation, etc.) and in machine learning (c.f. https://arxiv.org/abs/1906.10720) that use or require line attractors. The motivation of the paper would be much better tested on these tasks.

Minor comments
- The authors comment at the beginning of page 4 that by setting A=I, W=0, and h=0, the network contains a line attractor, but the more precise language would be to state that the network contains an N-dimensional plane attractor, where N is the number of units. Typically, 'line attractor' refers to a 1-dimensional manifold of fixed points along which the system can integrate inputs, but perturbations off of the line attractor are not remembered (decay back to the line attractor).

**Experience Assessment:**

I have published one or two papers in this area.

**Review Assessment: Checking Correctness Of Derivations And Theory:**

I carefully checked the derivations and theory.

**Review Assessment: Checking Correctness Of Experiments:**

I carefully checked the experiments.

**Review Assessment: Thoroughness In Paper Reading:**

I read the paper thoroughly.

---

> ### Author Response · Authors · 2019-11-06
> **point-by-point rebuttal**
>
> - Of course it is possible to analyze GRU and LSTM networks from a dynamical systems (DS) perspective, and of course they could be trained to exhibit line attractors, limit cycles, chaos, and all of that – no surprise, since after all, formally speaking, they are powerful discrete-time nonlinear DS themselves!
> Our point, however, is that they are much more tedious to analyze and understand than the PLRNN architecture we are propagating, for reasons we had given in the 3rd pg. of sect. 3.1: For the PLRNN we can track fixed points, limit cycles, and their stability *analytically*, and we can rather easily translate them into equivalent continuous time systems which brings additional advantages for analysis (see Suppl. 6.1.2); none of this is easily possible for LSTM or GRU!
> We are also not the first to point out the complexities of LSTM/GRU as a caveat to their training and analysis: In fact, to a large degree this motivated also the development of identity-initialized (e.g. Le, Jaitly, Hinton 2015) and orthogonal/ unitary RNN (e.g. Arjovsky, Shah, Bengio 2016) for solving long-dependency problems!
>
> - 'A' is a diagonal matrix (see below eq. 1), so off-diagonal elements *are* zero. The ‘true’ off-diagonal elements are in W, and these are indeed regularized toward zero (see eq. 4). Hence, the configuration is indeed driven toward forming a line attractor subspace (see also Suppl. Fig. S4).
> Besides, we show in our numerical experiments (Figs. 2 & 3) that the method *does work* and that its success is dependent on the regularization!
> As another note on the side, even if there would be a *perfect* line attractor subspace, it would still integrate external inputs (see eq. 1) and form an internal memory of them, and this memory could still be read out by the other units which receive inputs from this subspace!
>
> - First, please note that a line attractor is a continuous set of neutrally stable fixed points, not just a single fixed point (and hence not just a single Jacobian to be evaluated). More importantly, because our model is piecewise linear, all Jacobians in our case are strictly given by the matrices ‘A+W*D’ as defined in Suppl. sect. 6.1.2 & 6.1.3. Hence, our regularization is essentially a regularization of the Jacobians. If the regularization objective would be exactly met (as it will be for tau --> inf), there would be at least one eigenvalue exactly equal to 1. Please see also Suppl. Fig. S4.
> Also note that we have been careful to talk only about ‘line attractor dimensions’ – the overall dynamics *should not* be a line attractor, as this would rule out phenomena like chaos or limit cycles.
>
> - Except for vanilla RNN, we have included into our comparison only other methods that have been explicitly devised to exhibit long short-term memory properties, and hence these are the crucial comparisons in our minds. A simple L2 penalty on the weights would not convey any such mechanism, but of course it’s easy to do and we are happy to add it to the revised manuscript.
>
> - This last statement may reveal a fundamental misunderstanding about the aims of our work: The motivation of our method is *not* the study of line attractors (or systems based on them), we merely use line attractors as an effective mechanism for storing continuously valued variables in memory!! Figs. 2 & 3 moreover clearly demonstrate that it works – with this mechanism in place, we can actually outperform LSTMs on the addition and multiplication tasks; with this mechanism removed, performance breaks down!
> That we can successfully train our RNN even on challenging systems like the bursting neuron (3D, see 6.1.5), which the RNN then freely (!) reproduces when simulated, we think is actually a major feat (not a flaw), and something we have not quite seen elsewhere in the literature so far.
> Besides, recent proposals (see ref. above) for battling the vanishing/exploding gradient problem like orthogonal/unitary RNN or initialization-based approaches effectively use line attractors as well for their long-term memory (even LSTM and GRU essentially employ line attractors for their memory cells, at least that is what it comes down to in dynamical systems terms; i.e., linear recursive operations which map the values onto themselves). Our approach is clearly more along the lines of the recent orthogonal and identity-initialized RNNs that attempt to avoid the complexities of LSTM/GRU networks. However, orthogonal RNNs *cannot* produce many dynamical systems phenomena (like chaos; see e.g. Kerg et al. 2019), and this is what motivates our specific approach which is expressive enough to reproduce chaotic phenomena, multi-stability etc.
> We will attempt to make this clearer in the revision.
>
> Minor: For N>2 it is also not a plane anymore, that is why we used the term ‘line attractor dimensions’ initially, although, admittedly, we sometimes referred to this simply as ‘line attractors’ for short later in the text. We are happy to change this in the revision.

---

### Author Response · Authors · 2019-11-15
**Update on revisions made and summary of rebuttal**

Referee #3:
- We added a short discussion on EM vs. sequential VAE & SGVB to our Conclusions.

Referee #2:
- We added a sentence to sect. 4.1 on the relative performance of the rPLRNN vs. initialization-based approaches, and how we think the latter would further degrade in performance with increasing noise and task complexity.
- We performed runs with 100 units for iRNN and rPLRNN, and comparable # of parameters for LSTM, presented in new Suppl. Fig. S6, from which it is apparent that the results do not fundamentally change (in terms of ordering between models).
- We also reran numerical experiments for iRNN, rPLRNN, and LSTM with standard SGD. Again, the results point into the same direction, but for some of the models the number of stable runs we could obtain in the short time given is not sufficient yet (<=2) for inclusion into a paper intended for publication.
- We added a short discussion on EM vs. sequential VAE & SGVB to our Conclusions.
- We added some clarification on the form of the model prior and what is meant by ‘freely simulated’ below eq. 7.

Referee #1:
Unfortunately (for us) the first referee profoundly misinterpreted the basic aims of our study (most evident in 5th, but also in 1st and other comments) and also misread basic equations fundamental for understanding our approach (e.g. 2nd and 3rd comments; see our point-by-point reply). As a result, essentially each comment made was based on incorrect assumptions about the aims of our study, or our model framework and how it relates to dynamical properties of gated models (LSTM/GRU) (e.g. 1st or 5th comment). We furthermore believe this lack of understanding is not primarily our fault or simply due to bad writing, as the other two referees apparently found the text and equations clear enough. We are sorry that we have to point this out so clearly, but we feel somewhat pressed to do so, since based on his/her misreading this referee made a very strong judgment (rating of just ‘1’!) in a review process that will be publicly accessible with our names. This referee also made no further attempt to engage in discussions with us to clarify any of these misunderstandings. In the end, we felt there was little advice that could guide our revisions in this review which we perceived as rather ill-informed. We only included the requested L2-norm comparison in our revision (updated Fig. 2), which quite expectedly was one of the worst performing models.

Our manuscript demonstrates, for the first time as far as we are aware, how even very challenging nonlinear dynamical systems with widely differing time scales and tricky behavior (sharp spikes on top of slow oscillations) can be inferred from data such that their motion could be freely reproduced by the trained RNN, using, we had thought, a rather elegant and simple, but at the same time theoretically motivated, approach. It is quite disappointing that these achievements completely escaped this referee’s attention.

---

### Decision · Program_Chairs · 2019-12-19

**Decision:**

Reject

**Comment:**

The paper proposes  an interesting idea to leave a very simple form for piecewise-linear RNN, but separate units in to two types, one of which acts as memory. The "memory" units are penalized towards the linear attractor parameters, i.e. making elements of $A$ close to 1 and off-diagonal of $W$ close to $1$.
The benchmarks are presented that confirm the efficiency of the model.
The reviewer opinion were mixed; one "1", one "3" and one "6"; the Reviewer1 is far too negative and some of his claims are not very constructive, the "positive" reviewer is very short. Finally, the last reviewer raised a question about the actual quality on the results. This is not addressed. Although there is a motivation for such partial regularization, the main practical question is how many "memory neurons" are needed. I looked through the paper - this addressed only in the supplementary, where the value of $M_{reg}$ is mentioned (=0.5 M). For $M_{reg} = M$ it is the L2 penalty; what happens if the fraction is 0.1, 0.2, ... and more? A very crucial hyperparameter (and of course, smart selection of it can not be worse than L2RNN). This study is lacking. In my opinion, one can also introduce weights and sparsity constraints on them (in order to detect the number of "memory" neurons more-or less automatically). Although I feel this paper has a potential, it is not still ready for publication and could be significantly improved.